# Global Convergence of SGD For Logistic Loss on Two Layer Neural Nets

**Pulkit Gopalani**                                                                         *gopalani@umich.edu*
*Computer Science and Engineering*
*University of Michigan, Ann Arbor*

**Samyak Jha**                                                                         *samyakjha@iitb.ac.in*
*Department of Mathematics*
*Indian Institute of Technology, Bombay*

**Anirbit Mukherjee**                                                       *anirbit.mukherjee@manchester.ac.uk*
*Department of Computer Science*
*The University of Manchester*

**Reviewed on OpenReview:** *https:// openreview. net/ forum? id= 9TqAUYB6tC*

## Abstract

In this note, we demonstrate a first-of-its-kind provable convergence of SGD to the global minima of appropriately regularized logistic empirical risk of depth 2 nets – for arbitrary data with any number of gates with adequately smooth and bounded activations, like sigmoid and tanh, and for a class of distributions from which the initial weight is sampled. We also prove an exponentially fast convergence rate for continuous time SGD that also applies to smooth unbounded activations like SoftPlus. Our key idea is to show that the logistic loss function on any size neural net can be Frobenius norm regularized by a width-independent parameter such that the regularized loss is a "Villani function" – and thus be able to build on recent progress with analyzing SGD on such objectives.

## 1 Introduction

Modern developments in artificial intelligence have been significantly been driven by the rise of deep-learning. The highly innovative engineers who have ushered in this A.I. revolution have developed a vast array of heuristics that work to get the neural net to perform "human like" tasks. Most such successes, can mathematically be seen to be solving the function optimization/"risk minimization" question, $\min_{n \in \mathcal{N}} \mathbb{E}_{\boldsymbol{z} \in \mathcal{D}}[\ell(n, \boldsymbol{z})]$ where members of $\mathcal{N}$ are continuous functions representable by neural nets and $\ell : \mathcal{N} \times \mathrm{Support}(\mathcal{D}) \to [0, \infty)$ is called a "loss function" and the algorithm only has sample access to the distribution $\mathcal{D}$. The successful neural experiments can be seen as suggesting that there are many available choices of $\ell$, $\mathcal{N}$ & $\mathcal{D}$ for which highly accurate solutions to this seemingly extremely difficult question can be easily found. This is a profound mathematical mystery of our times

This work is about developing our understanding of some of the most ubiquitous methods of training nets. In particular, we shed light on how regularization can aid the analysis and help prove convergence to global minima for stochastic gradient methods for neural nets in hitherto unexplored and realistic parameter regimes.

In the last few years, there has been a surge in the literature on provable training of various kinds of neural nets in certain regimes of their widths or depths, or for very specifically structured data, like noisily realizable labels. Motivated by the abundance of experimental studies it has often been surmised that Stochastic Gradient Descent (SGD) on neural net losses – with proper initialization and learning rate – converges to a low–complexity solution, one that generalizes – when it exists (Zhang et al., 2018).

But, to the best of our knowledge a convergence result for any stochastic training algorithm applied to the logistic loss for even depth 2 nets (one layer of activations with any kind of non–linearity), without either an

assumption on the width or the data, has remained elusive so far. We recall that this is the most common way to train classifiers facing binary class labelled data.

In this work, we not only take a step towards addressing the above question in the theory of neural networks but we also do so while keeping to a standard algorithm, the Stochastic Gradient Descent (SGD). In light of the above, our key message can be summarily stated as follows,

**Theorem 1.1** (Informal Statement of Lemma 3.3). *If the initial weights are sampled from an appropriate class of distributions, then for nets with a single layer of sigmoid or tanh gates – for arbitrary data and size of the net – SGD on appropriately regularized logistic loss, while using constant steps of size $\mathcal{O}(\epsilon)$, will converge in $\mathcal{O}(\frac{1}{\epsilon})$ steps to weights at which the expected regularized loss would be $\epsilon$–close to its global minimum.*

We note that the threshold amount of regularization needed in the above *would be independent of the width of the nets.* Further, this threshold would be shown to scale s.t it can either naturally turn out to be proportionately small if the norms of the training data are small or can be made arbitrarily small by choosing outer layer weights to be small. Our above result is made possible by the crucial observation informally stated in the following lemma - which infact holds for more general nets than what is encompassed by the above theorem,

**Lemma 1.2.** *It is possible to add a constant amount of Frobenius norm regularization on the weights, to the logistic loss on depth-2 nets with activations like SoftPlus, sigmoid and tanh gates s.t with no assumptions on the data or the size of the net, the regularized loss would be a Villani function.*

Since our result stated above does not require any assumptions on the data, or the neural net width, we posit that this significantly improves on previous work in this direction. To the best of our knowledge, similar convergence guarantees in the existing literature either require some minimum neural net width – growing w.r.t. inverse accuracy and the training set size (NTK regime (Chizat et al., 2018; Du et al., 2018b)), infinite width (Mean Field regime (Chizat & Bach, 2018; Chizat, 2022; Mei et al., 2018)) or other assumptions on the data when the width is parametric (e.g. realizable data, (Ge et al., 2019; Zhou et al., 2021)).

In contrast to all these, we show that with appropriate $\ell_2$ regularization, SGD on logistic loss on 2–layer sigmoid / tanh nets converges to the global infimum of the loss. Our critical observation towards this proof is that the above standard losses on 2–layer nets – for a broad class of activation functions — are a "Villani function". Our proof get completed by leveraging the relevant results in Shi et al. (2020).

**Organization**   In Section 2 we shall give a literature review of existing proofs about guaranteed training of neural nets. In Section 3 we present our primary results – in particular Lemma 3.3 which uses special initialization of the weights to show the global convergence of SGD on regularized logistic loss with ±1 labelled data and and for gates like sigmoid and tanh. Additionally, in Theorem 3.5 we also point out that for our architecture, if using the SoftPlus activation, we can show that the underlying SDE converges in expectation to the global minimizer in linear time. In Section 4, we give a brief overview of the methods in Shi et al. (2020) and further details are given in Appendix D. In Section 3.2 we discuss some experiments with synthetic data, which show that there exist nets trained on the loss function considered such that they have high binary classification accuracy near the critical value of the regularizer considered for the proof. Further experiments with MNIST demonstrating similar phenomenon are given in Appendix E. We end in Section 6 with a discussion of various open questions that our work motivates. In Appendices A to C one can find the calculations needed in the main theorems' proofs.

## 2   Related Work

Firstly, we note that in recent times major advances have been made about understanding the statistical properties of doing binary classification by neural nets. In Zhou & Huo (2023), the authors consider {±1} labelled data distributed as a Gaussian Mixture Model and the labels satisfying a Tsybakov-type noise conditions with the noise exponent being $q$. The authors obtain that with probability $1 - \delta$ of sampling $n$ data, the difference between the population risk of the empirical risk minimizer and the Bayes' optimal

risk is upperbounded by $C_{q,d} \log\left(\frac{2}{\delta}\right)(\log n)^4\left(\frac{1}{n}\right)^{\frac{q+1}{q+2}}$, where $C_{q,d}$ is some constant depending on $q$ and data dimension $d$. To appreciate this, we note that earlier in Shen et al. (2022), similar bounds for CNNs with logistic loss were obtained. However, unlike the result in Shen et al. (2022), the excess risk bound obtained in Zhou & Huo (2023) for the hinge loss doesn't blow up with respect to the increasing smoothness of the minimizer of the risk over all measurable functions.

In the setting of finite–width neural nets trained on logistic loss for binary classification (Chatterji et al., 2021), it can be shown that if one has (a) small initial loss (poly $\left(\frac{1}{n}\right)$, where $n$ is the number of training data) and (b) 'smoothly approximately ReLU activation function', then the loss converges at a rate of $O\left(\frac{1}{t}\right)$ over $t$ steps of gradient descent. But to ensure the smallness of the initial loss, this result needs to assume a large width which scales polylogarithmically with inverse of the confidence parameter. In that limited sense this can be seen to be belonging to the larger framework of proofs at asymptotically wide nets which we review as follows.

**Review of the NTK Approach To Provable Neural Training :** One of the most popular parameter zones for theory of provable training of nets has been the so–called "NTK" (Neural Tangent Kernel) regime – where the width is a high degree polynomial in the training set size and inverse accuracy (a somewhat *unrealistic* regime) and the net's last layer weights are scaled inversely with width as the width goes to infinity. (Du et al., 2018a; Su & Yang, 2019; Allen-Zhu et al., 2019b; Du & Lee, 2018; Allen-Zhu et al., 2019a; Arora et al., 2019b; Li et al., 2019; Arora et al., 2019a; Chizat et al., 2018; Du et al., 2018b). The core insight in this line of work can be summarized as follows: for large enough width, SGD *with certain initializations* converges to a function that fits the data perfectly, with minimum norm in the RKHS defined by the neural tangent kernel – which gets specified entirely by the initialization (which is such that the initial output is of order one). A key feature of this regime is that the net's matrices do not travel outside a constant radius ball around the starting point – a property that is often not true for realistic neural net training scenarios.

In particular, for the case of depth 2 nets – with similarly smooth gates as we focus on – in Song et al. (2021) global convergence of gradient descent was shown using number of gates scaling sub-quadratically in the number of data - which, to the best of our knowledge, is the smallest known width requirement for such a convergence in a classification setup. On the other hand, for the special case of training depth 2 nets with ReLU gates on cross-entropy loss for doing binary classification, in Ji & Telgarsky (2020) it was shown that one needs to blow up the width only poly-logarithmically with target accuracy to get global convergence for SGD.

**Review of the Mean-Field Approach To Provable Neural Net Training :** In a separate direction of attempts towards provable training of neural nets, works like Chizat & Bach (2018) showed that a Wasserstein gradient flow limit of the dynamics of discrete time algorithms on shallow nets, converges to a global optimizer – if the convergence of the flow is assumed. We note that such an assumption is very non-trivial because the dynamics being analyzed in this setup is in infinite dimensions – a space of probability measures on the parameters of the net. Similar kind of non–asymptotic convergence results in this so–called 'mean–field regime' were also obtained.(Mei et al., 2018; Fang et al., 2021; Chizat & Bach, 2018; Nguyen & Pham, 2020; Sirignano & Spiliopoulos, 2022; Ren & Wang, 2022). The key idea in the mean–field regime is to replace the original problem of neural training which is a non-convex optimization problem in finite dimensions by a convex optimization problem in infinite dimensions – that of probability measures over the space of weights. The mean–field analysis necessarily require the probability measures (whose dynamics is being studied) to be absolutely–continuous and thus de facto it only applies to nets in the limit of them being infinitely wide.

We note that the results in the NTK regime hold without regularization while in many cases the mean–field results need it. (Mei et al., 2018; Chizat, 2022; Tzen & Raginsky, 2020).

In the next subsection we shall give a brief overview of some of the attempts that have been made to get provable deep-learning at parametric width.

**Need And Attempts To Go Beyond Large Width Limits of Nets** The essential proximity of the NTK regime to kernel methods and it being less powerful than finite nets has been established from multiple points of view. (Allen-Zhu & Li, 2019; Wei et al., 2019).

In He & Su (2020), the authors had given a a very visibly poignant way to see that the NTK limit is not an accurate representation of a lot of the usual deep-learning scenarios. Their idea was to define a notion of "local elasticity" – when doing a SGD update on the weights using a data say $\boldsymbol{x}$, it measures the fractional change in the value of the net at a point $\boldsymbol{x}'$ as compared to $\boldsymbol{x}$. It's easy to see that this is a constant function for linear regression - as is what happens at the NTK limit (Theorem 2.1 Lee et al. (2019)). But it has been shown in Dan et al. (2021) that this local-elasticity function indeed has non-trivial time-dynamics (particularly during the early stages of training) when a moderately large neural net is trained on logistic loss.

Specific to depth-2 nets – as we consider here – there is a stream of literature where analytical methods have been honed to this setup to get good convergence results without width restrictions - while making other structural assumptions about the data or the net. Janzamin et al. (2015) was one of the earliest breakthroughs in this direction and for the restricted setting of realizable labels they could provably get arbitrarily close to the global minima. For non-realizable labels they could achieve the same while assuming a large width but in all cases they needed access to the score function of the data distribution which is a computationally hard quantity to know. In a more recent development, Awasthi et al. (2021) have improved over the above to include ReLU gates while being restricted to the setup of realizable data and its marginal distribution being Gaussian.

One of the first proofs of gradient based algorithms doing neural training for depth$-2$ nets appeared in Zhong et al. (2017). In Ge et al. (2019) convergence was proven for training depth-2 ReLU nets for data being sampled from a symmetric distribution and the training labels being generated using a 'ground truth' neural net of the same architecture as being trained – the so-called "Teacher–Student" setup. For similar distributional setups, some of the current authors had in Karmakar et al. (2020) identified classes of depth$-2$ ReLU nets where they could prove linear-time convergence of training – and they also gave guarantees in the presence of a label poisoning attack. The authors in Zhou et al. (2021) consider another Teacher–Student setup of training depth 2 nets with absolute value activations. In this work, authors can get convergence in $\mathrm{poly}(d, \frac{1}{\epsilon})$ time, in a very restricted setup of assuming Gaussian data, initial loss being small enough, and the teacher neurons being norm bounded and 'well–separated' (in angle magnitude). Cheridito et al. (2022) get width independent convergence bounds for Gradient Descent (GD) with ReLU nets, however at the significant cost of having the restrictions of being only an asymptotic guarantee and assuming an affine target function and one–dimensional input data. While being restricted to the Gaussian data and the realizable setting for the labels, an intriguing result in Chen et al. (2021) showed that fully poly-time learning of arbitrary depth 2 ReLU nets is possible if one is in the "black-box query model".

**Related Work on Provable Training of Neural Networks Using Regularization**   Using a regularizer is quite common in deep-learning practice and in recent times a number of works have appeared which have established some of these benefits rigorously. In particular, Wei et al. (2019) show that for a specific classification task (noisy–XOR) definable in any dimension $d$, no NTK based 2 layer neural net can succeed in learning the distribution with low generalization error in $o(d^2)$ samples, while in $O(d)$ samples one can train the neural net using Frobenius/$\ell_2$–norm regularization. Nakkiran et al. (2021) show that for a specific optimal value of the $\ell_2$- regularizer the double descent phenomenon can be avoided for linear nets - and that similar tuning is possible even for real world nets.

In the seminal work Raginsky et al. (2017), it was pointed out that one can add a regularization to a gradient Lipschitz loss and make it satisfy the dissipativity condition so that Stochastic Gradient Langevin Dynamics (SGLD) provably converges to its global minima. But SGLD is seldom used in practice, and to the best of our knowledge it remains unclear if the observation in Raginsky et al. (2017) can be used to infer the same about SGD. Also it remains open if there exists neural net losses which satisfy all the assumptions needed in the above result. We note that the convergence time in Raginsky et al. (2017) for SGLD is $\mathcal{O}\left(\frac{1}{\epsilon^5}\right)$ using an $\mathcal{O}\left(\epsilon^4\right)$ learning rate, while in our Theorem 3.3 SGD converges in expectation to the global infimum of the regularized neural loss in time, $\mathcal{O}\left(\frac{1}{\epsilon}\right)$ using a $\mathcal{O}\left(\epsilon\right)$ step-length.

In summary, to the best of our knowledge, it has remained an unresolved challenge to show convergence of SGD for logistic loss on any neural architecture with a constant number of gates while not constraining the distribution of the data to a specific functional form. *In this work, we exploit the use of some regularization*

*to be able to resolve this optimization puzzle in our key result, Lemma 3.3 – and in our experiments we show that the regularization needed may not harm downstream classification performance. Thus we take a step towards bridging this important lacuna in the existing theory of stochastic optimization for neural nets in general.*

## 3  Setup and Main Results

We start with defining the neural net architecture, the loss function and the algorithm for which we will prove our convergence results.

**Definition 1** (**Constant Step-Size SGD On Depth-2 Nets**). *Let, $\sigma : \mathbb{R} \to \mathbb{R}$ (applied elementwise for vector valued inputs) be atleast once differentiable activation function. Corresponding to it, consider the width $p$, depth $2$ neural nets with fixed outer layer weights $\boldsymbol{a} \in \mathbb{R}^p$ and trainable weights $\boldsymbol{W} \in \mathbb{R}^{p \times d}$ as,*

$$\mathbb{R}^d \ni \boldsymbol{x} \mapsto f(\boldsymbol{x}; \boldsymbol{a}, \boldsymbol{W}) = \boldsymbol{a}^\top \sigma(\boldsymbol{W}\boldsymbol{x}) \in \mathbb{R}$$

*Then, corresponding to a given set of $n$ binary training data $(\boldsymbol{x}_i, y_i) \in \mathbb{R}^d \times \{+1, -1\}$, with $\|\boldsymbol{x}_i\|_2 \le B_x$, $i = 1, \ldots, n$ define the individual data logistic losses $\tilde{L}_i(\boldsymbol{W}) \coloneqq \log\left(1 + e^{-y_i f_i(W)}\right)$. Then for any $\lambda > 0$ let the regularized logistic empirical risk be,*

$$\tilde{L}(\boldsymbol{W}) \coloneqq \frac{1}{n} \sum_{i=1}^n \tilde{L}_i(\boldsymbol{W}) + \frac{\lambda}{2} \|\boldsymbol{W}\|_F^2 \tag{3.1}$$

*Correspondingly, we consider SGD with step-size $s > 0$ as,*

$$\boldsymbol{W}^{k+1} = \boldsymbol{W}^k - \frac{s}{b} \sum_{i \in \mathcal{B}_k} \nabla \tilde{L}_i(\boldsymbol{W}^k) - s\lambda \boldsymbol{W}^k$$

*where $\mathcal{B}_k$ is a randomly sampled mini-batch of size $b$.*

**Definition 2** (**Properties of the Activation $\sigma$**). *Let the $\sigma$ used in Definition 1 be bounded s.t. $|\sigma(x)| \le B_\sigma$, $C^\infty$, $L$–Lipschitz and $L'_\sigma$–smooth. Further assume that $\exists$ a constant vector $\boldsymbol{c}$ and positive constants $B_\sigma, M_D$ and $M'_D$ s.t $\sigma(\boldsymbol{0}) = \boldsymbol{c}$ and $\forall x \in \mathbb{R}, |\sigma'(x)| \le M_D, |\sigma''(x)| \le M'_D$ .*

In the framework of Shi et al. (2020), the required step-length for convergence of the above SGD and the corresponding measure of the rate of convergence of the loss depend on the gradient Lipschitz smoothness and the Poincaré constant of the Gibbs's measure of the loss function, respectively. Hence, towards stating the final results, in terms of the above constants we can now quantify these as follows,

**Lemma 3.1** (for Classification with Logistic Loss). *In the setup of binary classification as contained in Definition 3.1, and the definition $M_D$ and $L$ as given in Definition 2 above, there exists a constant $\lambda_c = \frac{M_D L B_x^2 \|a\|_2^2}{2}$ s.t $\forall \lambda > \lambda_c$ and $s > 0$ the Gibbs measure $\sim \exp\left(-\frac{2\tilde{L}}{s}\right)$ satisfies a Poincaré-type inequality with the corresponding constant $\lambda_s$.*

*Moreover, if the activation satisfies the conditions of Definition 2 then $\exists \mathrm{gLip}\left(\tilde{L}\right) > 0$ such that the empirical loss is $\mathrm{gLip}\left(\tilde{L}\right)$-smooth and we can bound the smoothness coefficient of the empirical loss as,*

$$\mathrm{gLip}(\tilde{L}) \le \sqrt{p} \left( \frac{\sqrt{p}\|\boldsymbol{a}\|_2 M_D^2 B_x}{4} + \left( \frac{2 + \|c\|_2 + \|\boldsymbol{a}\|_2 B_\sigma}{4} \right) M'_D B_x p + \lambda \right) \tag{3.2}$$

The precise form of the Poincaré-type inequalities used above is detailed in Theorem 4.1.

**Theorem 3.2** (**Bounds on Error for Arbitrary Initialization.**). *We continue in the setup of logistic loss from Definitions 1 and 2 and Lemma 3.1. For any $s > 0$, define the probability measure $\mu_s \coloneqq \frac{1}{Z_s} \exp\left(-\frac{2\tilde{L}(\boldsymbol{W})}{s}\right)$, $Z_s$ being the normalization factor. Then, $\forall\ T > 0$ and desired accuracy $\epsilon > 0$, $\exists$ constants $A(\tilde{L})$, $B(T, \tilde{L})$ and $C(s, \tilde{L})$ s.t if the above SGD is executed at a constant step-size*

$$s = s^*(\epsilon, T) \coloneqq \min\left(\frac{1}{\mathrm{gLip}(\tilde{L})}, \epsilon \cdot \frac{1}{(A(\tilde{L}) + B(T, \tilde{L}))}\right)$$

*with the weights $\boldsymbol{W}^0$ initialized from any distribution with p.d.f $\rho_{\mathrm{initial}} \in L^2(\frac{1}{\mu_{s^*}})$ and then, the error at the end of having taken $k = \frac{T}{s^*}$ SGD steps can bounded as,*

$$\mathbb{E}\tilde{L}(\boldsymbol{W}^k) - \inf_{\boldsymbol{W}} \tilde{L}(\boldsymbol{W}) \le \epsilon + C(s^*, \tilde{L}) \|\rho_{\mathrm{initial}} - \mu_{s^*}\|_{\mu_{s^*}^{-1}} e^{-s^* \lambda_{s^*} \cdot k}$$

*where*

$$\|\rho_{\mathrm{initial}} - \mu_{s^*}\|_{\mu_{s^*}^{-1}} \coloneqq \left(\int_{\mathbb{R}} \left(\rho_{\mathrm{initial}}(x) - \mu_{s^*}\right)^2 \mu_{s^*}^{-1} dx\right)^{\frac{1}{2}}$$

*measures the 'gap' between the initial distribution and the stationary distribution*

**Lemma 3.3** (**Global Convergence of SGD on Sigmoid and Tanh Neural Nets of 2 Layers for Any Width and Any Data - for Binary Classification With Logistic Loss**). *We continue in the setup of logistic loss from Definitions 1 and 2 and Lemma 3.1. For any $s > 0$, define the probability measure $\mu_s \coloneqq \frac{1}{Z_s} \exp\left(-\frac{2\tilde{L}(\boldsymbol{W})}{s}\right)$, $Z_s$ being the normalization factor. Then, $\forall\ T > 0$, and desired accuracy, $\epsilon > 0$, $\exists$ constants $A(\tilde{L})$, $B(T, \tilde{L})$ and $C(s, \tilde{L})$ s.t if the above SGD is executed at a constant step-size*

$$s = s^*\left(\frac{\epsilon}{2}, T\right) \coloneqq \min\left(\frac{1}{\mathrm{gLip}(\tilde{L})}, \frac{\epsilon}{2 \cdot (A(\tilde{L}) + B(T, \tilde{L}))}\right)$$

*with the weights $\boldsymbol{W}^0$ initialized from a distribution with p.d.f $\rho_{\mathrm{initial}} \in L^2(\frac{1}{\mu_{s^*}})$ and $\|\rho_{\mathrm{initial}} - \mu_{s^*}\|_{\mu_{s^*}^{-1}} \le \frac{\epsilon}{2 \cdot C(s^*, \tilde{L})} \cdot e^{\lambda_{s^*} \cdot T}$ – then, in expectation, the regularized empirical risk of the net, $\tilde{L}$ would converge to its global infimum, with the rate of convergence given as,*

$$\mathbb{E}\tilde{L}(\boldsymbol{W}^{\frac{T}{s^*}}) - \inf_{\boldsymbol{W}} \tilde{L}(\boldsymbol{W}) \le \epsilon.$$

The proof of Theorem 3.2 is given in Section 5 and the proof of Lemma 3.1 can be read off from the calculations done as a part of the proof of Theorem 3.2. Lemma 3.3 can be obtained along the same lines as Theorem 3.2 as follows.

Similar to Theorem 3.2, we consider SGD executed at a constant step size $s = s^*(\frac{\epsilon}{2}, T)$ – and as earlier this constant is defined in terms of the constants $A(L), B(T, \tilde{L}), C(s, \tilde{L})$. Further, we invoke the condition, that we consider the initial weights of the SGD being sampled from an initial distribution with p.d.f $\rho_{\mathrm{initial}}$ such that,

$$\|\rho_{\mathrm{initial}} - \mu_{s^*}\|_{\mu_{s^*}^{-1}} \le \frac{\epsilon}{2} \cdot \frac{e^{\lambda_{s^*} \cdot T}}{C(s^*, \tilde{L})}$$

Then for $k = \frac{T}{s^*}$, combining the above into Theorem 3 from Shi et al. (2020) (as was also used to get the guarantee of Theorem 3.2) we get the guarantee in Lemma 3.3,

$$\mathbb{E}\tilde{L}(\boldsymbol{W}^k) - \inf_{\boldsymbol{W}} \tilde{L}(\boldsymbol{W}) \le \frac{\epsilon}{2} + C(s^*, \tilde{L}) \|\rho_{\mathrm{initial}} - \mu_{s^*}\|_{\mu_{s^*}^{-1}} e^{-s^* \lambda_{s^*} \cdot k} \le \frac{\epsilon}{2} + \frac{\epsilon}{2} = \epsilon$$

We make a few quick remarks about the nature of the above guarantees, *Firstly,* we note that the "time horizon" $T$ above is a free parameter - which in turn parameterizes the choice of the step-size and the initial weight distribution. Choosing a larger $T$ makes the constraints on the initial weight distribution weaker at the cost of making the step-size smaller and the required number of SGD steps larger. But for any value of $T$, the above theorem guarantees that SGD, initialized from weights sampled from a certain class of distributions, converges in expectation to the global minima of the regularized empirical loss for our nets for any data and width, in time $\mathcal{O}(\frac{1}{\epsilon})$ using a learning rate of $\mathcal{O}(\epsilon)$.

*Secondly,* we note that the phenomenon of a lower bound on the regularization parameter being needed for certain nice learning theoretic property to emerge has been seen in kernel settings too. (Yang et al., 2017).

Also, to put into context the emergence of a critical value of the regularizer for nets as in the above theorem, we recall the standard result, that there exists an optimal value of the $\ell_2$−regularizer at which the excess risk of the similarly penalized linear regression becomes dimension free (Proposition 3.8, Bach (2022)). However, we recall that the quantities required for computing this "optimal" regularizer are not knowable while training and hence it is not practically implementable. Thus, we see that for binary classification, one can define a notion of an "optimal" regularizer and it remains open to investigate if such a similar threshold of regularization also exists for nets. Our above theorem can be seen as a step in that direction.

*Thirdly,* we note that the lowerbounds on training time of neural nets proven in works like Goel et al. (2020) do not apply here since these are proven for SQ algorithms and SGD is not of this type.

*Finally,* note that the threshold values of regularization computed above, $\lambda_c$, do not explicitly depend on the training data or the neural architecture, consistent with observations in Anthony & Bartlett (2009); Zhang et al. (2021). It depends on the activation and scales with the norm of the input data and the outer layer of weights.

For intuition, suppose in the binary classification setting we set the outer layer weights s.t we have $\|\boldsymbol{a}\|_2 \cdot B_x = 1$. This leads to $\lambda_c = \frac{M_D L}{2}$. For the sigmoid activation, $\sigma_\beta(x)$, by calculations as above we would get, $\lambda_c$ in this case (say $\lambda_c^{si,\beta}$) to be $= \frac{\beta^2}{32}$. Since $\beta = 1$ is the most widely used setting for the above sigmoid activation, this results in,

$$\lambda_c^{si,1} \approx 0.03125 \tag{3.3}$$

### 3.1 Global Convergence of Continuous Time SGD on Nets with SoftPlus Gates

In, Shi et al. (2020) the authors had established that, if the loss $\tilde{L}$ is gradient Lipschitz, then over any fixed time horizon $T > 0$, as $s \to 0$, the dynamics of the SGD in Definition 1 is arbitrarily well approximated (in expectation) by the unique global solution that exists for the Stochastic Differential Equation (SDE),

$$d\mathbf{W}_s(t) = -\nabla \tilde{L}(\mathbf{W}_s(t))\,dt + \sqrt{s}\,d\mathbf{B}(t) \qquad \text{(SGD–SDE)} \tag{3.4}$$

where $\mathbf{B}(t)$ is the standard Brownian motion. The SGD convergence proven in the last section critically uses this mapping to a SDE. In Shi et al. (2020) it was further pointed out that if we only want to get a non-asymptotic convergence rate for the continuous time dynamics, the smoothness of the loss function is not needed and only the Villani condition suffices. In this short section we shall exploit this to show convergence of continuous time SGD on $\tilde{L}$ with the activation function being the unbounded 'SoftPlus'. Also, in contrast to the guarantee about SGD in the previous subsection here we shall see that the SDE converges exponentially faster i.e *at a linear rate.*

**Definition 3** (SoftPlus activation)**.** *For $\beta > 0$, $x \in \mathbb{R}$, define the SoftPlus activation function as*

$$\text{SoftPlus}_\beta(x) = \frac{1}{\beta} \log_e (1 + \exp(\beta x))$$

**Remark.** *Note that $\lim_{\beta \to \infty} \text{SoftPlus}_\beta(x) = \text{ReLU}(x)$. Also note that for $f(x) = SoftPlus_\beta(x)$, $f'(x) = \sigma_\beta(x)$ (sigmoid function as defined above) and hence $|f'(x)| \le M_D$ for $M_D = 1$ and $f(x)$ is $L$−Lipschitz for $L = 1$.*

Recall the following fact that was proven as a part of Lemma 3.1,

**Lemma 3.4.** *There exists a constant* $\lambda_c \coloneqq \frac{M_D L B_x^2 \|\boldsymbol{a}\|_2^2}{2}$ *s.t* $\forall$ $\lambda > \lambda_c$ & $s > 0$, *the Gibbs' measure* $\mu_s \coloneqq \frac{1}{Z_s} \exp\left(-\frac{2\tilde{L}(\boldsymbol{W})}{s}\right)$, $Z_s$ *being the normalization factor, satisfies a Poincaré-type inequality with the corresponding constant* $\lambda_s$.

**Theorem 3.5** (**Continuous Time SGD Converges to Global Minima of SoftPlus Nets in Linear Time**). *We consider the SDE as given in Equation 4.1 on a Frobenius norm regularized logistic empirical loss on depth−2 neural nets as specified in Equation 3.1, while using* $\sigma(x) = \text{SoftPlus}_\beta(x)$ *for* $\beta > 0$, *the regularization threshold being s.t* $\lambda > \lambda_c = \frac{M_D L B_x^2 \|\boldsymbol{a}\|_2^2}{2}$ *and with the weights* $\boldsymbol{W}_0$ *being initialized from a distribution with p.d.f* $\rho_{\text{initial}} \in L^2(\frac{1}{\mu_s})$.

*Then, for any* $S > 0$, $\exists$ $G(S, \tilde{L})$ *and* $C(s, \tilde{L})$, *an increasing function of* $s$, *s.t for any step size* $0 < s \le \min\left\{\frac{\epsilon}{2G(S,\tilde{L})}, S\right\}$ *and for* $t \ge \frac{1}{\lambda_s} \log\left(\frac{2\,C(s,\tilde{L})\|\rho_{\text{initial}} - \mu_s\|_{\mu_s^{-1}}}{\epsilon}\right)$ *we have that,*

$$\mathbb{E}\,\tilde{L}(\boldsymbol{W}(t)) - \min_{\boldsymbol{W}} \tilde{L}(\boldsymbol{W}) \le \epsilon.$$

*Proof.* The SoftPlus function is Lipschitz, hence using the same analysis as in (Section 5), we can claim that for $\lambda > \lambda_c$ the loss function in Definition 1 with SoftPlus activations is a Villani function (and hence confining, by definition).

Then, from Proposition 3.1 of Shi et al. (2020) it follows that, $\exists\, C(s, \tilde{L})$, an increasing function of $s$, that satisfies,

$$\left|\mathbb{E}\tilde{L}(\boldsymbol{W}_s(t)) - \mathbb{E}\tilde{L}(\boldsymbol{W}_s(\infty))\right| \le C(s, \tilde{L})\|\rho_{\text{initial}} - \mu_s\|_{\mu_s^{-1}}\, e^{-\lambda_s t}.$$

From Proposition 3.2 of Shi et al. (2020) it follows that, for any $S > 0$, for $s \in (0, S)$, $\exists\, G(S, \tilde{L})$ that quantifies the excess risk at the stationary point of the SDE as,

$$\tilde{L}(\boldsymbol{W}(\infty)) - \min_{\boldsymbol{W}} \tilde{L} \le G(S, \tilde{L})\, s$$

Combining the above, the final result claimed follows as in Corollary 3.3 in Shi et al. (2020). $\square$

## 3.2 An Experimental Demonstration of the Maintenance of Classification Accuracy At Various Regularizations at Different Widths

For further illustration of the ramifications of the novel convergence theorems shown above, in here we present some experimental studies of doing binary classification by training depth 2 sigmoid activated nets with the regularized loss considered in the above convergence proofs. And we will be using the normalizations that correspond to the theoretically needed threshold value of the regularizer being $\lambda_c = 0.03125$ (Equation 3.3).

We sampled the data from a clearly separable dataset with a margin − $n$ data vectors in $d$−dimensions were sampled as a $n \times d$ normally distributed matrix whereby after sampling each row vector was normalized to have unit norm. The data whose last coordinate were $> 0.2$ were assigned label $+1$ and where the last coordinate was $< -0.2$ was assigned label $-1$ and the rest of the data were discarded. In our experimental setting, we fixed to $d = 10$. The data were split into being 20% test data and the rest used for training.

Then we simulate SGD based training on the above data for multiple neural nets at various values of $\lambda$ in $[0, \lambda_C]$ and for neural net widths $p$. The step-length in the experiments is constant across all widths and lambda, across all settings.

The elements of the (trainable) weight Matrix $\boldsymbol{W}_0$ of dimension $p \times d$ were initialized from a standard normal distribution, i.e $\boldsymbol{W}_0 \sim \mathcal{N}(0, \boldsymbol{I}_{p \times d})$. Likewise, the elements of the (fixed) outer layer $\boldsymbol{a}$, of dimension $1 \times p$ were sampled as $\boldsymbol{a} \sim \mathcal{N}(0, \boldsymbol{I}_{1 \times p})$ and then normalized.

For all experimental settings the neural networks were trained for 500 epochs, and with a mini-batch size of 256. At the end of training we measured the test accuracy of classification - as the downstream metric of measuring the goodness of training.

As shown in the experimental graph (Figure 1), we demonstrate examples of nets trained on regularized logistic loss showing remarkable accuracy for regularization $\lambda \leq \lambda_c$ – even though the model was not been exposed to the $0-1$ criteria during training.

The code for the experiments can be found at this Colaboratory File.

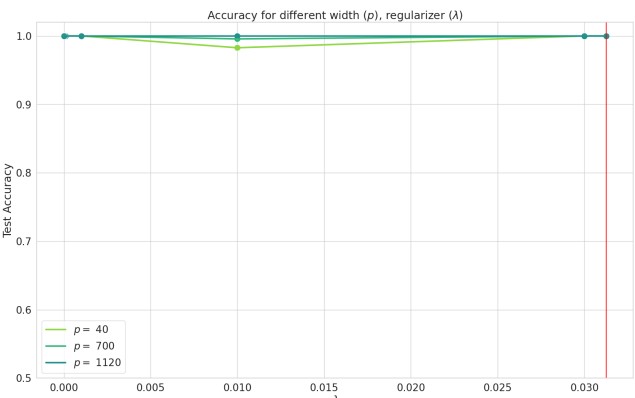

Figure 1: Test Accuracy across various widths $p$ and regularizer $\lambda$

Thus we have demonstrated that the threshold amount of regularization that was needed for the proof of convergence may not at all harm the downstream performance metric of classification.

For further insights, the reader can see Appendix E, where additional experiments on the MNIST dataset are detailed. These experiments showcase that the regularized loss function considered in the aforestated theorems can be trained via SGD to achieve remarkable classification performance on real data too.

## 4 Overview of Shi et al. (2020)

In Section 5, we will give the proof for our Theorem 3.2. As relevant background for the proof, in here we shall give a brief overview of the framework in Shi et al. (2020), which can be summarized as follows : suppose one wants to minimize the function $\tilde{L}(\boldsymbol{W}) \coloneqq \frac{1}{n} \sum_{i=1}^{n} \tilde{L}_i(\boldsymbol{W})$, where $i$ indexes the training data, $\boldsymbol{W}$ is in the parameter space (the optimization space) of the loss function and $\tilde{L}_i$ is the loss evaluated on the $i^{th}$–datapoint. On this objective, a constant step-size mini-batch implementation of the Stochastic Gradient Descent (SGD) consists of doing the following iterates, $\boldsymbol{W}_{k+1} = \boldsymbol{W}_k - \frac{s}{b} \sum_i \nabla \tilde{L}_i(\boldsymbol{W}_k)$, where the sum is over a mini-batch (a randomly sampled subset of the training data) of size $b$ and $s$ is the fixed step-length. In, Shi et al. (2020) the authors established that over any fixed time horizon $T > 0$, as $s \to 0$, the dynamics of this SGD is arbitrarily well approximated (in expectation) by the Stochastic Differential Equation (SDE),

$$\mathrm{d}\boldsymbol{W}_s(t) = -\nabla \tilde{L}(\boldsymbol{W}_s(t)) \, \mathrm{d}t + \sqrt{s} \, \mathrm{d}\boldsymbol{B}(t) \qquad \text{(SGD–SDE)} \tag{4.1}$$

where $\boldsymbol{B}(t)$ is the standard Brownian motion. We recall that the Markov semigroup operator $P_t$ for a stochastic process $X_t$ and its infinitesimal generator $\mathcal{L}$ are given as, $P_t f(x) \coloneqq \mathbb{E}[f(X_t) \mid X_0 = x]$ and $\mathcal{L}f \coloneqq \lim_{t \downarrow 0} \frac{P_t f - f}{t}$.

Invoking the Forward Kolmogorov equation $\partial_t f = \mathcal{L}^* f$, one obtains the following Fokker–Planck–Smoluchowski PDE governing the evolution of the density of the SDE,

$$\frac{\partial \rho_s}{\partial t} = \langle \nabla \rho_s, \nabla \tilde{L} \rangle + \rho_s \Delta \tilde{L} + \frac{s}{2} \Delta \rho_s \qquad \text{(FPS)} \tag{4.2}$$

Further, under appropriate conditions on $\tilde{L}$ the above implies that the density $\rho_s(t)$ converges exponentially fast to the Gibbs' measure corresponding to the objective function i.e the distribution with p.d.f

$$\mu_s = \frac{1}{Z_s}\exp\left(-\frac{2\tilde{L}(\boldsymbol{W})}{s}\right) \tag{4.3}$$

where $Z_s$ is the normalization factor. The sufficient conditions on $\tilde{L}$ that were shown to be needed to achieve this "mixing" and to know a rate for it, are that of $\tilde{L}$ be a "Villani Function" as defined below,

**Definition 4** (**Villani Function** (Villani (2009); Shi et al. (2020)))**.** *A map $f : \mathbb{R}^d \to \mathbb{R}$ is called a Villani function if it satisfies the following conditions,*

*1. $f \in C^\infty$*

*2. $\lim_{\|\boldsymbol{x}\|\to\infty} f(\boldsymbol{x}) = +\infty$*

*3. $\int_{\mathbb{R}^d} \exp\left(-\frac{2f(\boldsymbol{x})}{s}\right)\mathrm{d}\boldsymbol{x} < \infty \ \forall s > 0$*

*4. $\lim_{\|\boldsymbol{x}\|\to\infty}\left(-\Delta f(\boldsymbol{x}) + \frac{1}{s}\cdot\|\nabla f(\boldsymbol{x})\|^2\right) = +\infty \ \forall s > 0$*

*Further, any $f$ that satisfies conditions 1 – 3 is said to be "confining".*

From Lemma 5.2 Shi et al. (2020), the empirical or the population risk, $\tilde{L}$, being confining is sufficient for the FPS PDE (equation 4.2) to evolve the density of SGD–SDE (equation 4.1) to the said Gibbs' measure.

But, to get non-asymptotic guarantees of convergence – even for the SDE (Corollary 3.3, Shi et al. (2020)), we need a Poincaré–type inequality to be satisfied (as defined below) by the aforementioned Gibbs' measure $\mu_s$. A sufficient condition for this Poincaré–type inequality to be satisfied is if a confining loss function $\tilde{L}$ also satisfied the last condition in definition 4 (and is consequently a Villani function).

**Theorem 4.1** (Poincaré–type Inequality (Shi et al. (2020)))**.** *Given a $f : \mathbb{R}^d \to \mathbb{R}$ which is a Villani Function (Definition 4), for any given $s > 0$, define a measure with the density, $\mu_s(\boldsymbol{x}) = \frac{1}{Z_s}\exp\left(-\frac{2f(\boldsymbol{x})}{s}\right)$, where $Z_s$ is a normalization factor. Then this (normalized) Gibbs' measure $\mu_s$ satisfies a Poincare-type inequality i.e $\exists \ \lambda_s > 0$ (determined by $f$) s.t $\forall h \in C_c^\infty(\mathbb{R}^d)$ we have,*

$$\mathrm{Var}_{\mu_s}[h] \le \frac{s}{2\lambda_s}\cdot\mathbb{E}_{\mu_s}\left[\|\nabla h\|^2\right]$$

The reader can see Appendix D for a more detailed sketch of the key proof in Shi et al. (2020) about why this SGD-SDE considered above mixes to a Gibbs's distribution for objectives being the Villani function

The approach of Shi et al. (2020) has certain key interesting differences from many other contemporary uses of SDEs to prove the convergence of discrete time stochastic algorithms. Instead of focusing on the convergence of parameter iterates $\boldsymbol{W}^k$, they consider the dynamics of the expected error i.e $\mathbb{E}[\tilde{L}(\boldsymbol{W}^k)]$, for $\tilde{L}$ being the empirical or the population risk. This leads to a transparent argument for the convergence of $\mathbb{E}[\tilde{L}(\boldsymbol{W}^k)]$ to $\inf_{\boldsymbol{W}} \tilde{L}(\boldsymbol{W})$, by leveraging standard results which help one pass from convergence guarantees on the SDE to a convergence of the SGD.

We note that Shi et al. (2020) achieve this conversion of guarantees from SDE to SGD by additionally assuming gradient smoothness of $\tilde{L}$ – and we would show that this assumption holds for the natural neural net loss functions that we consider.

## 5 Proof of Theorem 3.2

*Proof.* Note that $\tilde{L}$ being a confining function can be easily read off from Definition 4. Further, as shown in Appendix A, the following inequalities hold,

$$\|\nabla_{\boldsymbol{W}} L\left(\boldsymbol{W}\right)\|^2 \ge \left(\lambda^2 - \frac{\lambda\|\boldsymbol{a}\|_2^2 M_D B_x^2 L}{2}\right)\|\boldsymbol{W}\|_F^2 - \lambda\|\boldsymbol{W}\|_F\|\boldsymbol{a}\|_2 M_D B_x \left(1 + \frac{\|\boldsymbol{a}\|_2\|\boldsymbol{c}\|_2}{2}\right)$$

$$\Delta_{\boldsymbol{WW}}\tilde{L} \le p\left[M_{\tilde{d}}^2 B_x^2 \|\boldsymbol{a}\|_2^2 + \|\boldsymbol{a}\|_2 \left[\left(B_y + \|\boldsymbol{a}\|_2 \left(\|\boldsymbol{c}\|_2 + L B_x\|\boldsymbol{W}\|_F\right)\right)\left(M_D' B_x^2\right)\right] + \lambda d\right] \qquad (5.1)$$

Combining the above two inequalities we can conclude that, $\exists$ functions $g_1, g_2, g_3$ such that,

$$\frac{1}{s}\left\|\nabla_{\boldsymbol{W}}\tilde{L}\right\|^2 - \Delta_{\boldsymbol{WW}}\tilde{L} \ge g_1(\lambda, s)\|\boldsymbol{W}\|_F^2 - g_2(\lambda, s)\|\boldsymbol{W}\|_F + g_3(\lambda, s)$$

where in particular,

$$g_1(\lambda, s) = \lambda^2 - 2\lambda \cdot M_D L B_x^2 \|\boldsymbol{a}\|_2^2.$$

Hence we can conclude that for $\lambda > \lambda_c \coloneqq 2 M_D L B_x^2 \|\boldsymbol{a}\|_2^2$, $\forall s > 0$, $\frac{1}{s}\left\|\nabla_{\boldsymbol{W}}\tilde{L}\right\|^2 - \Delta_{\boldsymbol{WW}}\tilde{L}$ diverges as $\|\boldsymbol{W}\| \to +\infty$, since $g_1(\lambda, s) > 0$. The key aspect of the above analysis being that the bound on $\Delta_{\boldsymbol{WW}}$ does not depend on $\|\boldsymbol{W}\|_F^2$. Thus we have, that the following limit holds,

$$\lim_{\|\boldsymbol{W}\|_F \to +\infty} \left(\frac{1}{s}\left\|\nabla_{\boldsymbol{W}}\tilde{L}\right\|^2 - \Delta_{\boldsymbol{WW}}\tilde{L}\right) = +\infty$$

for the range of $\lambda$ as given in the theorem, hence proving that $\tilde{L}$ is a Villani function.

Towards getting an estimate of the step-length as given in the theorem statement, we also show in Appendix B that the loss function $\tilde{L}$ is gradient–Lipschitz with the smoothness coefficient being upperbounded as,

$$\mathrm{gLip}(\tilde{L}) \le \sqrt{p}\left(\frac{\sqrt{p}\|\boldsymbol{a}\|_2 M_D^2 B_x}{4} + \left(\frac{2 + \|\boldsymbol{c}\|_2 + \|\boldsymbol{a}\|_2 B_\sigma}{4}\right) M_D' B_x p + \lambda\right)$$

Now we can invoke Theorem 3 (Part 1), Shi et al. (2020) with appropriate choice of $s$, as given in the theorem statement to get the result as given in Theorem 3.2. In Appendix C one can find a discussion of the computation of the specific constants involved in the expression for the suggested step-length $s^\star$.  $\square$

## 6  Conclusion

To the best of our knowledge, in this work, we have shown the first proof of convergence of SGD to the global minima of logistic loss on a neural net whilst not making any assumptions about the data or the width of the net. Our result relied on the convergence of discrete-time algorithms like SGD to their continuous time counterpart (SDEs) - a theme that has lately been an active field of research.

We recall that Ji & Telgarsky (2020) is among the most related works to this as it shares with us the same SGD algorithm, primary loss function, neural architecture, and data labels. Their proof exploits specific properties of the (leaky) ReLU like positively homogeneous activation function and is limited to asymptotically wide networks. In contrast, our convergence result (Lemma 3.3) applies to activation functions like tanh and sigmoid not covered by them and most importantly accommodates arbitrary data and network widths. But to achieve these flexibilities we needed to exploit a parametric (in data norm and last layer norm) amount of regularization and impose certain constraints on the distribution from which the initial weights are sampled.

By juxtaposing these two results, multiple intriguing research directions for the immediate future are revealed - which can significantly advance our understanding of the provable training of neural networks. Specifically, we advocate for further investigation into reformulating the constraints on the initial weight distribution in more intuitive terms. It could also be promising to explore whether natural weight initialization schemes adhere to the sufficient criteria we require. We also point out that experiments suggest that convergence as we prove can happen at lower values of regularization (possibly in a data-dependent way) and hence that suggests trying for tighter analysis to show that neural losses can be Villani functions.

We posit that trying to reproduce our Theorem 3.3 using a direct analysis of the dynamics of SGD could be a fruitful venture leading to interesting insights. Our results also motivate a new direction of pursuit in

deep-learning theory, centered around understanding the nature of the Poincaré constant of Gibbs' measures induced by neural net losses.

Our experiments further demonstrated that there exists neural nets and binary class labelled data such that optimizing on our provably good smooth loss functions also does highly accurate classification. This motivates a fascinating direction of future pursuit about the phenomenon of classification calibration or Bayes consistency i.e to be able to analytically identify cases where convergence guarantees as given here could be extended to guarantees on the classification error.

Lastly, given the new method of proving neural training by SGD that has been initiated in this work, it naturally begets the question if these proof techniques could also resolve similar mysteries for more exotic loss functions and neural architectures that are in use.

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

## A Towards Establishing the Villani condition for the Empirical Logistic Loss

We start with the following observation,

**Lemma A.1.** *Letting $\boldsymbol{w}_j$ denote the $j^{\text{th}}$ row of $\boldsymbol{W}$ we have,*

$$\nabla_{\boldsymbol{w}_j} f_i(\boldsymbol{W}) = a_j \sigma'\left(\boldsymbol{w}_j^\top \boldsymbol{x}_i\right) \boldsymbol{x}_i$$

In the following, $\|\boldsymbol{W}\|$ for a matrix $\boldsymbol{W}$ denotes its spectral or operator norm. We recall that the regularized logistic loss for training the given neural net on data $\mathbb{D} = \{(\boldsymbol{x}_i, y_i)\}_{i=1}^n$ is,

$$\tilde{L}(\boldsymbol{W}) := \frac{1}{n} \sum_{i=1}^n \tilde{L}_i(\boldsymbol{W}) + R_\lambda(\boldsymbol{W}) := \frac{1}{n} \sum_{i=1}^n \left\{ \frac{1}{2}\left(y_i - f(\boldsymbol{x}_i; \boldsymbol{a}, \boldsymbol{W})\right)^2 \right\} + \frac{\lambda}{2}\|\boldsymbol{W}\|_F^2$$

Explicitly stated, the SGD iterates with step-length $s > 0$ that we analyze on the above loss are,

$$\begin{aligned}
\boldsymbol{W}^{k+1} &= \boldsymbol{W}^k - s\left[\frac{1}{b}\sum_{i \in \mathcal{B}_k} \nabla \tilde{L}_i(\boldsymbol{W}^k) + \nabla R_\lambda(\boldsymbol{W}^k)\right] \\
&= \boldsymbol{W}^k - \frac{s}{b}\sum_{i \in \mathcal{B}_k}\left(-\left(y_i - f(\boldsymbol{x}_i; \boldsymbol{a}, \boldsymbol{W}^k)\right) \cdot \nabla_{\boldsymbol{W}^k} f(\boldsymbol{x}_i; \boldsymbol{a}, \boldsymbol{W}^k)\right) - s\lambda \boldsymbol{W}^k \\
&= (1 - s\lambda)\boldsymbol{W}^k + \frac{s}{b}\sum_{i \in \mathcal{B}_k}\left[(y_i - f(\boldsymbol{x}_i; \boldsymbol{a}, \boldsymbol{W}^k)) \cdot \nabla_{\boldsymbol{W}^k} f(\boldsymbol{x}_i; \boldsymbol{a}, \boldsymbol{W}^k)\right]
\end{aligned}$$

We also note the following observation,

**Lemma A.2.**

$$\frac{1}{1 + e^{y_i f_i(\boldsymbol{W})}} \leq \frac{1}{2} + \frac{|f_i(\boldsymbol{W})|}{4} \tag{A.1}$$

*Proof.*

$$\frac{1}{1 + e^{y_i f_i(\boldsymbol{W})}} \leq \frac{1}{1 + e^{-|f_i(\boldsymbol{W})|}} = \frac{e^{|f_i(\boldsymbol{W})|}}{1 + e^{|f_i(\boldsymbol{W})|}}$$

Consider the function $g(z) = \frac{e^z}{1+e^z}, \ z \in [0, \infty)$

$$g(0) = \frac{1}{2}, \ \ g'(z) = \frac{e^z}{(1 + e^z)^2} \leq \frac{1}{4}$$

Hence, we have

$$g(z) - g(0) = \int_0^z g'(z) \leq \int_0^z \frac{1}{4} = \frac{z}{4}$$

Therefore,

$$g(z) \leq \frac{1}{2} + \frac{z}{4} \tag{A.2}$$

$\square$

**Lower-bounding the Norm of the Gradient of the Empirical Logistical Loss**

Recall that we are using the following empirical loss function,

$$L(\boldsymbol{W}) = \frac{1}{n}\sum_{i=1}^{n}\ell(y_i f_i(\boldsymbol{W})) + \frac{\lambda}{2}\|\boldsymbol{W}\|_2^2 \tag{A.3}$$

where $\ell(z) = \log(1 + e^{-z})$ is the logistic loss function. It follows that,

$$\nabla_{\boldsymbol{w}_j} L(\boldsymbol{W}) = \frac{1}{n}\sum_{i=1}^{n}\ell'(y_i f_i(\boldsymbol{W}))\, y_i \nabla_{\boldsymbol{w}_j} f(\boldsymbol{W}) + \lambda \boldsymbol{w}_j \tag{A.4}$$

Therefore,

$$\nabla_{\boldsymbol{w}_j} L(\boldsymbol{W}) = \frac{1}{n}\sum_{i=1}^{n}\frac{-y_i}{1 + e^{y_i f_i(\boldsymbol{W})}}\nabla_{\boldsymbol{w}_j} f(\boldsymbol{W}) + \lambda \boldsymbol{w}_j$$

And that implies,

$$\begin{aligned}
\left\|\nabla_{\boldsymbol{w}_j} L(\boldsymbol{W})\right\|_2^2 &= \lambda^2\|\boldsymbol{w}_j\|_2^2 - 2\left\langle \lambda\boldsymbol{w}_j, \frac{1}{n}\sum_{i=1}^{n}\frac{y_i}{1 + e^{y_i f_i(\boldsymbol{W})}}\nabla_{\boldsymbol{w}_j} f(\boldsymbol{W})\right\rangle + \left\|\frac{1}{n}\sum_{i=1}^{n}\frac{-y_i}{1 + e^{y_i f_i(\boldsymbol{W})}}\nabla_{\boldsymbol{w}_j} f(\boldsymbol{W})\right\|^2 \\
&\geq \lambda^2\|\boldsymbol{w}_j\|_2^2 - 2\left\langle \lambda\boldsymbol{w}_j, \frac{1}{n}\sum_{i=1}^{n}\frac{y_i}{1 + e^{y_i f_i(\boldsymbol{W})}}\nabla_{\boldsymbol{w}_j} f(\boldsymbol{W})\right\rangle \\
&\geq \lambda^2\|\boldsymbol{w}_j\|_2^2 - 2\lambda\|\boldsymbol{w}_j\|_2\left(\frac{1}{n}\sum_{i=1}^{n}\frac{\left\|\nabla_{\boldsymbol{w}_j} f_i(\boldsymbol{W})\right\|}{1 + e^{y_i f_i(\boldsymbol{W})}}\right) \\
&\geq \lambda^2\|\boldsymbol{w}_j\|_2^2 - 2\lambda\|\boldsymbol{w}_j\|_2\left(\frac{1}{n}\sum_{i=1}^{n}\frac{|a_j|\left|\sigma'(\boldsymbol{w}_j^\top \boldsymbol{x}_i)\right|\|\boldsymbol{x_i}\|_2}{1 + e^{y_i f_i(\boldsymbol{W})}}\right)
\end{aligned} \tag{A.5}$$

In the last line above we have invoked Lemma A.1. Now invoking Lemma A.2 and recalling the definition of $M_D$ and $B_x$ in the above, we can claim that,

$$\begin{aligned}
\left\|\nabla_{\boldsymbol{w}_j} L(\boldsymbol{W})\right\|_2^2 &\geq \lambda^2\|\boldsymbol{w}_j\|_2^2 - 2\lambda\|\boldsymbol{w}_j\|_2|a_j|M_D\|\boldsymbol{x}_i\|\left(\frac{1}{n}\sum_{i=1}^{n}\frac{1}{2} + \frac{|f_i(\boldsymbol{W})|}{4}\right) \\
&\geq \lambda^2\|\boldsymbol{w}_j\|_2^2 - 2\lambda\|\boldsymbol{w}_j\|_2|a_j|M_D B_x\left(\frac{1}{2} + \frac{\|\boldsymbol{a}\|_2(LB_x\|\boldsymbol{W}\|_2 + \|\boldsymbol{c}\|_2)}{4}\right)
\end{aligned}$$

Summing the above over all $j$ and using Cauchy-Schwartz inequality, $\sum_{j=1}^{p}\|\boldsymbol{w}_j\|_2 \cdot |a_j| \leq \|\boldsymbol{W}\|_F \cdot \|\boldsymbol{a}\|$ and $\|\boldsymbol{W}\|_2 \leq \|\boldsymbol{W}\|_F$, we get,

$$\|\nabla_{\boldsymbol{W}} L(\boldsymbol{W})\|^2 \geq \left(\lambda^2 - \frac{\lambda\|\boldsymbol{a}\|_2^2 M_D B_x^2 L}{2}\right)\|\boldsymbol{W}\|_F^2 - \lambda\|\boldsymbol{W}\|_F\|\boldsymbol{a}\|_2 M_D B_x\left(1 + \frac{\|\boldsymbol{a}\|_2\|\boldsymbol{c}\|_2}{2}\right) \tag{A.6}$$

**Analyzing the Laplacian of the Empirical Logistic Loss**

We begin with observing that,

$$\begin{aligned}
\left|\nabla_{\boldsymbol{w}_j} \cdot (\nabla_{\boldsymbol{w}_j} L(\boldsymbol{W}))\right| &= \left|\nabla_{\boldsymbol{w}_j} \cdot \left(\frac{1}{n}\sum_{i=1}^{n}\ell'(y_i f_i(\boldsymbol{W}))\, y_i \nabla_{\boldsymbol{w}_j} f(\boldsymbol{W})\right) + \lambda d\right| \\
&\leq \left|\frac{1}{n}\sum_{i=1}^{n}\ell''(f_i(\boldsymbol{W}))\left\|\nabla_{\boldsymbol{w}_j} f(\boldsymbol{W})\right\|_2^2\right| + \left|\frac{1}{n}\sum_{i=1}^{n}\ell'(y_i f_i(\boldsymbol{W}))\, y_i \Delta_{\boldsymbol{w}_j} f(\boldsymbol{W})\right| + \lambda d
\end{aligned}$$

In above we have defined, $\Delta_{\boldsymbol{w}_j} f = \nabla_{\boldsymbol{w}_j} \cdot (\nabla_{\boldsymbol{w}_j} f)$ We recall that, $\ell''(z) = \frac{e^z}{(1+e^z)^2} \leq \frac{1}{4}$, as show in Lemma A.2. Further by invoking Lemma A.1, the definition of $M_D$ and $B_x$ we have,

$$
\begin{aligned}
\left| \frac{1}{n} \sum_{i=1}^{n} \ell''\left(y_i f_i\left(\boldsymbol{W}\right)\right) \left\|\nabla_{\boldsymbol{w}_j} f\left(\boldsymbol{W}\right)\right\|_2^2 \right| &\leq \left| \frac{1}{n} \sum_{i=1}^{n} \frac{1}{4} \left\|\nabla_{\boldsymbol{w}_j} f\left(\boldsymbol{W}\right)\right\|_2^2 \right| \\
&\leq \left| \frac{1}{n} \sum_{i=1}^{n} \frac{1}{4} |a_j|^2 (\sigma'(\boldsymbol{w}_j^\top x_i))^2 \|\boldsymbol{x}_i\|^2 \right| \\
&\leq \frac{M_D^2 B_x^2 \|\boldsymbol{a}\|_2^2}{4}
\end{aligned}
\tag{A.7}
$$

Recalling the definition of $M_D'$ we have,

$$
\begin{aligned}
\left| \frac{1}{n} \sum_{i=1}^{n} \ell'\left(y_i f_i\left(\boldsymbol{W}\right)\right) y_i \Delta_{\boldsymbol{w}_j} f\left(\boldsymbol{W}\right) \right| &\leq \left| \frac{1}{n} \sum_{i=1}^{n} \left(\frac{1}{2} + \frac{|f_i\left(\boldsymbol{W}\right)|}{4}\right) y_i \Delta_{\boldsymbol{w}_j} f\left(\boldsymbol{W}\right) \right| \\
&\leq \left( \frac{2 + \|c\|_2}{4} + \frac{\|\boldsymbol{a}\|_2 L B_x \|\boldsymbol{W}\|_F}{4} \right) B_x^2 M_D' \|\boldsymbol{a}\|_2
\end{aligned}
\tag{A.8}
$$

And hence,

$$
\left| \nabla_{\boldsymbol{w}_j} \cdot \left( \nabla_{\boldsymbol{w}_j} L\left(\boldsymbol{W}\right) \right) \right| \leq \left( \frac{2 + \|c\|_2}{4} + \frac{\|\boldsymbol{a}\|_2 L B_x \|\boldsymbol{W}\|_F}{4} \right) B_x^2 M_D' \|\boldsymbol{a}\|_2 + \frac{M_D^2 B_x^2 \|\boldsymbol{a}\|_2^2}{4} + \lambda d
\tag{A.9}
$$

Summing over all j, we get,

$$
\Delta_{\boldsymbol{W}} L\left(\boldsymbol{W}\right) \leq \sum_{j=1}^{p} \left| \nabla_{\boldsymbol{w}_j} \cdot \left( \nabla_{\boldsymbol{w}_j} L\left(\boldsymbol{W}\right) \right) \right| \leq p \left[ \left( \frac{2 + \|c\|_2}{4} + \frac{\|\boldsymbol{a}\|_2 L B_x \|\boldsymbol{W}\|_F}{4} \right) B_x^2 M_D' \|\boldsymbol{a}\|_2 + \frac{M_D^2 B_x^2 \|\boldsymbol{a}\|_2^2}{4} + \lambda d \right]
\tag{A.10}
$$

## B  Bounding the Gradient Lipschitzness Coefficient of the Empirical Logistic Loss

Towards the upcoming computation, we define the following function,

$$
g_j\left(\boldsymbol{W}\right) \coloneqq \nabla_{\boldsymbol{w}_j} L = \frac{1}{n} \sum_{i=1}^{n} \ell'\left(y_i f_i\left(\boldsymbol{W}\right)\right) y_i \nabla_{\boldsymbol{w}_j} f_i\left(\boldsymbol{W}\right) + \lambda \boldsymbol{w}_j
\tag{B.1}
$$

Consequently, corresponding to two values of the weight matrix $\boldsymbol{W}_1$ and $\boldsymbol{W}_2$, we obtain that,

$$
\begin{aligned}
&\|g_j\left(\boldsymbol{W}_2\right) - g_j\left(\boldsymbol{W}_1\right)\|_2 = \\
&\left\| \frac{1}{n} \sum_{i=1}^{n} y_i \left( \ell'\left(y_i f_i\left(\boldsymbol{W}_2\right)\right) \nabla_{\boldsymbol{w}_{2,j}} f_i\left(\boldsymbol{W}_2\right) - \ell'\left(y_i f_i\left(\boldsymbol{W}_1\right)\right) \nabla_{\boldsymbol{w}_{1,j}} f_i\left(\boldsymbol{W}_1\right) \right) + \lambda \boldsymbol{w}_{2,j} - \lambda \boldsymbol{w}_{1,j} \right\|_2 \\
&\leq \left\| \frac{1}{n} \sum_{i=1}^{n} y_i \left( \ell'\left(y_i f_i\left(\boldsymbol{W}_2\right)\right) \nabla_{\boldsymbol{w}_{2,j}} f_i\left(\boldsymbol{W}_2\right) - \ell'\left(y_i f_i\left(\boldsymbol{W}_1\right)\right) \nabla_{\boldsymbol{w}_{1,j}} f_i\left(\boldsymbol{W}_1\right) \right) \right\|_2 + \|\lambda \boldsymbol{w}_{2,j} - \lambda \boldsymbol{w}_{1,j}\|_2 \\
&\leq \frac{B_x \|\boldsymbol{a}\|_2}{n} \sum_{i=1}^{n} \left| \ell'\left(y_i f_i\left(\boldsymbol{W}_2\right)\right) \sigma'(\boldsymbol{w}_{2,j}^\top \boldsymbol{x}_i) - \ell'\left(y_i f_i\left(\boldsymbol{W}_1\right)\right) \sigma'(\boldsymbol{w}_{1,j}^\top \boldsymbol{x}_i) \right| + \|\lambda \boldsymbol{w}_{2,j} - \lambda \boldsymbol{w}_{1,j}\|_2
\end{aligned}
\tag{B.2}
$$

In the last line above we have invoked Lemma A.1 and the fact that $y_i \in \{1, -1\}$ for $i = 1, \ldots, n$. Hence, the problem simplifies to finding the Lipschitz constant of $\ell'\left(y_i f_i\left(\boldsymbol{W}\right)\right) \sigma'(\boldsymbol{w}_j^\top \boldsymbol{x}_i)$ Define $h_k$ as follows:

$$
\boldsymbol{h}_k(\boldsymbol{W}) \coloneqq \nabla_{\boldsymbol{w}_k} \left( \ell'\left(y_i f_i\left(\boldsymbol{W}\right)\right) \sigma'(\boldsymbol{w}_j^\top \boldsymbol{x}_i) \right)
\tag{B.3}
$$

$$\|\boldsymbol{h}_k(\boldsymbol{W})\|_2 = \left\|\nabla_{\boldsymbol{w}_k}\left(\ell'\left(y_i f_i(\boldsymbol{W})\right)\sigma'(\boldsymbol{w}_j^\top \boldsymbol{x}_i)\right)\right\|_2$$

$$= \left\|a_k \ell''\left(y_i f_i(\boldsymbol{W})\right)\sigma'(\boldsymbol{w}_j^\top \boldsymbol{x}_i)\sigma'(\boldsymbol{w}_k^\top \boldsymbol{x}_i)\boldsymbol{x}_i + \mathbf{1}_{k=j}\ell'\left(y_i f_i(\boldsymbol{W})\right)\sigma''(\boldsymbol{w}_j^\top \boldsymbol{x}_i)\boldsymbol{x}_i\right\|_2$$

$$\leq \left\|a_k \ell''\left(y_i f_i(\boldsymbol{W})\right)\sigma'(\boldsymbol{w}_j^\top \boldsymbol{x}_i)\sigma'(\boldsymbol{w}_k^\top \boldsymbol{x}_i)\boldsymbol{x}_i\right\| + \left\|\mathbf{1}_{k=j}\ell'\left(y_i f_i(\boldsymbol{W})\right)\sigma''(\boldsymbol{w}_j^\top \boldsymbol{x}_i)\boldsymbol{x}_i\right\|_2$$

$$\leq \frac{\|\boldsymbol{a}\|_2 M_D^2 B_x}{4} + \left\|\mathbf{1}_{k=j}\left(\frac{1}{2} + \frac{|f_i(\boldsymbol{W})|}{4}\right)\sigma''(\boldsymbol{w}_j^\top \boldsymbol{x}_i)\boldsymbol{x}_i\right\|_2$$

$$\leq \frac{\|\boldsymbol{a}\|_2 M_D^2 B_x}{4} + \left(\frac{2 + \|c\|_2}{4} + \frac{\|\boldsymbol{a}\|_2 B_\sigma}{4}\right)M_D' B_x \sqrt{p} = L_{\mathrm{prod}}$$

where in the second term the $\sqrt{p}$ factor comes in from using Cauchy-Schwarz inequality. We concatenate these functions along the indices $k = 1, 2, \ldots, p$, to get

$$\boldsymbol{h}(\boldsymbol{W}) \coloneqq \nabla_{\boldsymbol{W}}\left[f(\boldsymbol{x}_i; \boldsymbol{a}, \boldsymbol{W})\sigma'(\boldsymbol{w}_j^\top \boldsymbol{x}_i)\right] = [\boldsymbol{h}_1(\boldsymbol{W}), \boldsymbol{h}_2(\boldsymbol{W}), \ldots, \boldsymbol{h}_p(\boldsymbol{W})]$$

And hence,

$$\|\boldsymbol{h}(\boldsymbol{W})\|_2 \leq \sqrt{p} L_{\mathrm{prod}}$$

Therefore, the Lipschitz constant of $\ell'\left(y_i f_i(\boldsymbol{W})\right)\sigma'(\boldsymbol{w}_j^\top \boldsymbol{x}_i)$ is,

$$\frac{\|\boldsymbol{a}\|_2 M_D^2 B_x \sqrt{p}}{4} + \left(\frac{2 + \|c\|_2}{4} + \frac{\|\boldsymbol{a}\|_2 B_\sigma}{4}\right)M_D' B_x p$$

Hence the Lipschitz constant of $g_j(\boldsymbol{W})$ is

$$\frac{\|\boldsymbol{a}\|_2^2 M_D^2 B_x^2 \sqrt{p}}{4} + \left(\frac{2 + \|c\|_2}{4} + \frac{\|\boldsymbol{a}\|_2 B_\sigma}{4}\right)M_D' B_x^2 \|\boldsymbol{a}\|_2 p + \lambda$$

Proceeding as in the case of $\boldsymbol{h}(\boldsymbol{W})$ above, we now concatenate the above gradients $\boldsymbol{g}_j$ w.r.t the index $j$ in a vector form (of dimension $pd$) to get the following $pd$–dimensional gradient vector of the empirical loss,

$$\nabla_{\boldsymbol{W}}\tilde{L} = \boldsymbol{g}(\boldsymbol{W}) \coloneqq [\boldsymbol{g}_1(\boldsymbol{W}), \boldsymbol{g}_2(\boldsymbol{W}), \ldots, \boldsymbol{g}_p(\boldsymbol{W})]$$

and the Lipschitz constant of $\boldsymbol{g}$ - and hence the gradient Lipschitz constant for $\tilde{L}$ to be bounded as,

$$\mathrm{gLip}(\tilde{L}) \leq \sqrt{p}\left(\frac{\sqrt{p}\|\boldsymbol{a}\|_2 M_D^2 B_x}{4} + \left(\frac{2 + \|c\|_2}{4} + \frac{\|\boldsymbol{a}\|_2 B_\sigma}{4}\right)M_D' B_x p + \lambda\right)$$

Thus we get the expression as required in the proof in Section 5.

## C  Defining the Constants $C(s, \tilde{L})$ and $\lambda_s$ of Theorem 3.2

Invoking Theorem 3 from Shi et al. (2020) with a "time horizon" parameter $T > 0$, $f = \tilde{L}$, the convergence guarantee for running $k$ steps of SGD at a step-size $s$ as given in Theorem 3.2 for $k \cdot s \in (0, T)$ and $s \in (0, 1/\mathrm{gLip}(\tilde{L}))$ can be given as,

$$\mathbb{E}\tilde{L}(\boldsymbol{W}^k) - \inf_{\boldsymbol{W}}\tilde{L}(\boldsymbol{W}) \leq (A(\tilde{L}) + B(T, \tilde{L}))s + C(s, \tilde{L})\|\rho - \mu_s\|_{\mu_s^{-1}}e^{-s\lambda_s k} \tag{C.1}$$

where $A, B, C$ are constants as mentioned in Shi et al. (2020). In particular, the $C$ therein was defined as follows

$$C(s, \tilde{L}) = \left(\int_{\mathbb{R}^{p \times d}}(\tilde{L}(\boldsymbol{W}) - \min_{\boldsymbol{W}}\tilde{L}(\boldsymbol{W}))^2 \mu_s(\boldsymbol{W})\,d\boldsymbol{W}\right)^{1/2}$$

for $\mu_s(\boldsymbol{W})$ being the Gibbs' measure, $\mu_s(\boldsymbol{W}) = \frac{1}{Z_s}\exp\left(-\frac{2\tilde{L}(\boldsymbol{W})}{s}\right)$ with $Z_s$ being the normalization factor.

For determining $\lambda_s$, Shi et al. (2020) consider the function $V_s(\boldsymbol{W}) = \|\nabla\tilde{L}\|^2/s - \Delta\tilde{L}$. Let $R_{0,s} > 0$ be large enough such that $V_s(\boldsymbol{W}) > 0$ for $\|\boldsymbol{W}\|_F \geq R_{0,s}$. For $R_s > R_{0,s}$, Shi et al. (2020) define $\epsilon(R_s)$ as

$$\epsilon(R_s) = \frac{1}{\inf\{V_s(\boldsymbol{W}) : \|\boldsymbol{W}\|_F \geq R_s\}}$$

where $R_s$ is assumed large enough such that $\int_{\|\boldsymbol{W}\|_F \leq R_s} d\mu_s \geq 1/2$. For $B_{R_s}$ as the ball of radius $R_s$ centered at origin in $\mathbb{R}^{p\times d}$, Shi et al. (2020) define

$$\mu_{s,R_s} = \left[\int_{\|\boldsymbol{W}\|_F \leq R_s} d\mu_s(\boldsymbol{W})\right]^{-1} \mu_s(\boldsymbol{W})\mathbf{1}_{\|\boldsymbol{W}\|_F \leq R_s}.$$

Using the Poincaré inequality in a bounded domain[Evans (2010), Theorem 1, Chapter 5.8], Shi et al. (2020) define the constant $C(R_s)$ to be s.t the the following holds $\forall h \in C_c^\infty(\mathbb{R}^d)$,

$$\int_{\boldsymbol{W}\in\mathbb{R}^{p\times d}} h^2 d\mu_{s,R_s} \leq s \cdot C(R_s) \int_{\boldsymbol{W}\in\mathbb{R}^{p\times d}} \|\nabla h\|^2 \mu_{s,R_s} d\mu_{s,R_s} + \left(\int_{\boldsymbol{W}\in\mathbb{R}^{p\times d}} h\, d\mu_{s,R_s}\right)^2$$

Then the key quantity $\lambda_s$ occurring in the aforementioned convergence guarantee for SGD was shown to be,

$$\lambda_s = \frac{1 + 3s\left(\inf_{\boldsymbol{W}\in\mathbb{R}^{p\times d}} V_s(\boldsymbol{W})\right)\epsilon(R_s)}{2(C(R_s) + 3\epsilon(R_s))}$$

## D  Further Details of The Key Idea in Shi et al. (2020) About the Convergence of the SGD-SDE (Equation 4.1)

In this appendix we outline the steps of the key proof in Shi et al. (2020) which prove that that solution $\boldsymbol{W}_s(t)$ of the SGD-SDE (Equation 4.1) has a well-controlled bound on the error it makes at any time $t$ in terms of how far away it is from the global infimum of the objective function $\tilde{L}$.

**Theorem D.1** ((Main) Theorem 1 of Shi et al. (2020)). *If $\tilde{L}$ is both confining and Villani, $\exists\lambda_s > 0$ for any $s > 0$ ("learning rate") s.t for a certain functions $\epsilon(s,\tilde{L}) > 0$ which is strictly increasing in $s$ and $D(s,\boldsymbol{W},\rho) \geq 0$ where $\rho$ is the initial distribution, we have expotential convergence of the expected excess risk as,*

$$\mathbb{E}\tilde{L}(\boldsymbol{W}_s(t)) - \inf_{\boldsymbol{z}} \tilde{L}(\boldsymbol{z}) \leq \epsilon(s) + D(s,\boldsymbol{W},\rho)e^{-\lambda_s t}$$

Towards outlining the proof of the above we will denote $\mathbb{E}[\tilde{L}(\boldsymbol{W}_s(\infty))] \coloneqq \mathbb{E}_{\boldsymbol{W}\sim\mu_s}\tilde{L}(\boldsymbol{W})$, where $\mu_s$ is as given in equation 4.3 . This is justified by the following convergence theorem - which we count as the first of the 3 main steps to be taken to prove Theorem D.1.

### Step I

**Theorem D.2** (Lemma 5.2 in Shi et al. (2020)). *If $\tilde{L}$ satisfies the confining condition and if the initial distribution of the S.D.E is $\rho \in L^2(\frac{1}{\mu_s})$ then the unique solution $\rho_s(t) \in C^1([0,\infty), L^2(\frac{1}{\mu_s}))$ of the F.P.S differential equation (4.2) of the S.D.E, $\mathrm{d}\boldsymbol{W}_s = -\nabla\tilde{L}(\boldsymbol{W}_s)\,\mathrm{d}t + \sqrt{s}\,\mathrm{d}\boldsymbol{W}$ converges in $L^2(\frac{1}{\mu_s})$ to the Gibbs' invariant distribution $\mu_s$.*

Now we note the following decomposition of error that is considered in Theorem D.1.

$$\mathbb{E}\tilde{L}(\boldsymbol{W}_s(t)) - \inf_{\boldsymbol{z}} \tilde{L}(\boldsymbol{z}) = \left(\mathbb{E}\tilde{L}(\boldsymbol{W}_s(t)) - \mathbb{E}[\tilde{L}(\boldsymbol{W}_s(\infty))]\right) + \left(\mathbb{E}[\tilde{L}(\boldsymbol{W}_s(\infty))] - \inf_{\boldsymbol{z}} \tilde{L}(\boldsymbol{z})\right)$$

Next we bound each of the 2 terms in the RHS above via the following two theorems.

**Step II**

**Theorem D.3** (**Proposition 3.1 of** Shi et al. (**2020**))**.** *For $\tilde{L}$ being confining and Villani and for any $s > 0$ $\exists \lambda_s > 0$ s.t we have a function $C(s, \tilde{L}) > 0$ (an increasing function in $s$) s.t $\forall t \geq 0$ we have,*

$$\left| \mathbb{E}\tilde{L}(\boldsymbol{W}_s(t)) - \mathbb{E}\tilde{L}(\boldsymbol{W}_s(\infty)) \right| \leq C(s, \tilde{L}) \cdot e^{-\lambda_s t} \cdot \left\| \rho - \mu_s \right\|_{\frac{1}{\mu_s}}$$

**Step III**

**Theorem D.4** (**Proposition 3.2 of** Shi et al. (**2020**))**.** *The excess risk at stationarity $\epsilon(s)$ is s.t for all $S > 0$ and $s \in (0, S]$, $\exists A(S, \tilde{L})$ s.t,*

$$\epsilon(s) = \mathbb{E}[\tilde{L}(\boldsymbol{W}_s(\infty))] - \tilde{L}^* \leq A(S, \tilde{L}) \cdot s$$

Combining the above two theorems (in Step II and Step III respectively) we are led to the key upperbound stated in Theorem D.1, with an appropriate definition of the $D$ function therein.

## E  Experimental Demonstration of Maintenance of Classification Accuracy on MNIST Dataset

For further illustration, we present experimental studies performing binary classification between pairs of digits from the MNIST dataset by training depth-2, sigmoid activated nets and regularized with the coefficient being at the threshold where the loss was proven to become a Villani function.

We do the binary classification experiments on the digit pairs $(0, 1)$ and $(2, 7)$. In our experiments, the elements of the trainable weight matrix $\boldsymbol{W}_0$ (of dimension $12 \times 784$) is initialized from the standard normal distribution and so is the fixed outer layer of dimensions $1 \times 12$ – which was then rescaled by the largest data norm for the value of $\lambda_c$ to be as given in equation 3.3.

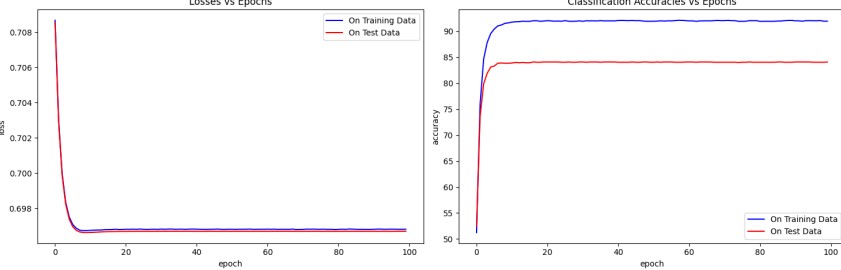

Figure 2: Batch Size $= 3000$, $\lambda = \lambda_c = 0.03125$ and the net being trained has 12 sigmoid gates

**Experiment on Digits 2 and 7**  This is the experiment shown in Figure 2, In this case the model was trained for 100 epochs with a mini-batch size of 3000. We measured the test accuracy of classification - as the downstream metric of measuring the goodness of training and at the end of training, we achieved an accuracy of $\approx 84\%$.

**Experiment on Digits 0 and 1**  This is the experiment shown in Figure 3. Here the model was trained for 100 epochs with a mini-batch size of 3000. We measured the test accuracy of classification as the downstream metric of measuring the goodness of training and at the end of training, we achieved $\approx 90\%$ accuracy

Thus we have experimentally demonstrated that the threshold amount of regularization that was needed for the proof of convergence may not at all harm the downstream performance metric of classification for even real data. In both the cases above we demonstrated examples of nets being trained on logistic loss for regularization parameter being $\lambda = \lambda_c$ and achieving good accuracy on the classification metric even though the model was not exposed to the $0 - 1$ criteria during training.

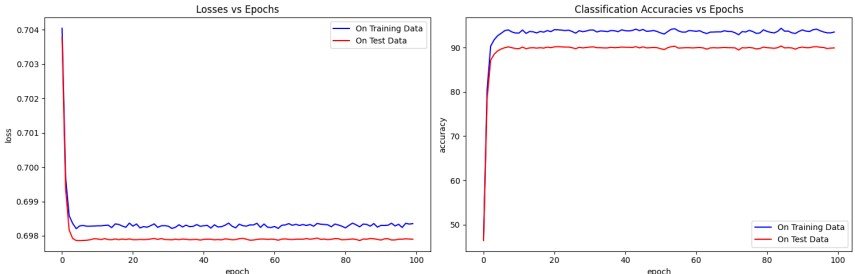

Figure 3: Batch Size = 3000, $\lambda = \lambda_c = 0.03125$ and the net being trained has 12 sigmoid gates.

Codes for these experiments can be found at this link for the $(2, 7)$ classification experiment and at this link for the $(0, 1)$ classification experiment

