# OpenReview forum: "Global Convergence of SGD For Logistic Loss on Two Layer Neural Nets"
_TMLR — Accepted by TMLR_

### Review · Reviewer_tgay · 2023-10-25

**Summary Of Contributions:**

This paper studies the problem of training a two-layer neural network with SGD on the logistic loss function. By relating the SGD dynamics in the small step-size regime to the limiting SDE and verifying that the objective potential satisfies a Poincar\'e inequality, a local convergence result for SGD is established which is independent of the width of the network and the input distribution.

**Audience:**

No

**Broader Impact Concerns:**

Not applicable.

**Claims And Evidence:**

No

**Requested Changes:**

I believe that the current version of the manuscript might not be ready for publication, and would suggest the following:

## Major Comments:
* I am not sure if the $O(1/\epsilon)$ convergence rate is correct if $\lambda_s$ can depend on $s = O(\epsilon)$. It would be a very interesting contribution to provide explicit estimates for $\lambda_s$, hence achieving explicit convergence rates, especially in a setting where the optimization problem is still non-convex. Such a contribution would also distinguish this work from Shi et al., 2020 at a technical level.

* It would greatly help if the authors could add a discussion on why or why not standard (Gaussian) initializations are able to achieve the initialization requirement of Theorem 3.2 / Lemma 3.3.

* Another interesting contribution would be to show that for some data distributions, the amount of regularization $\lambda$ required does not hurt performance. Such a result (theoretical or experimental) would be more interesting if the distribution is such that there is a gap between the performance of linear models and neural networks.

## Minor Comments:
There is some room for improving the writing style of the manuscript, for example:
* It is not clear why Theorem 3.2 is stated as a theorem and the closely related Lemma 3.3 which has a broader result is stated as a lemma. It might make more sense to present Lemma 3.3 as a Theorem and Theorem 3.2 as its corollary.

* The introduced mini-batch size is never used in the rest of the paper.

* Lemma 3.1 seems a bit out of place and perhaps the readers would need additional context to understand what is the significance of this lemma (hence appearing right before presenting the main theorem) and its relation to the main results.

**Strengths And Weaknesses:**

# Strengths:
The analysis can accommodate any width and input distribution.

# Weaknesses:
* While the manuscript provides a good summary of its proof technique, in its current form, there might not be a lot of new aspects to learn from it. Specifically, the arguments mostly directly follow those of Shi et al., 2020, and the main contribution seems to be verifying that a certain Villani condition holds in the investigated setting through standard calculations.

* The dependence of $\lambda_s$ on $s$ is not known. As a result, we cannot obtain a convergence rate, and in particular, it is not clear without additional discussion on $\lambda_s$ whether the claimed convergence rate of $O(1/\epsilon)$ is correct, since $s$ itself depends on $\epsilon$.

* It is not theoretically clear if the regime of regularization $\lambda$ chosen in the manuscript is sufficiently small to allow good classification performance. The authors have verified this experimentally, although in an arguably simple setting where linear predictors also achieve 100% accuracy.

* It is not discussed whether standard (Gaussian) initialization methods can achieve $\rho_\mathrm{initial} \in L^2(1/\mu_{s^*})$, as such a condition might require a relatively fast tail decay for small $s^*$. Furthermore, dimension dependencies are hidden in the bounds.

* There is some room for improving the writing of the paper, with examples provided below.

---

> ### Author Response · Authors · 2023-12-18
> **New Experiments Added and Result Statements Modified for Clarity**
>
> We thank the reviewer for their careful reading of the paper. Kindly find below the specific clarifications that we would like to offer for the very pertinent issues that you have raised.
>
> > I am not sure if the $O(\frac{1}{\epsilon})$ convergence rate is correct...
>
> We want to clarify that at no point in our paper do we assert that run-time is universally ${\cal O}(\frac{1}{\epsilon})$ for achieving $\epsilon$ accurate training.
>
> Secondly, we would like to point out, that in the (newly numbered) Lemma 3.3, it is shown that the run-time is ${\cal O}(\frac{1}{\epsilon})$ contingent upon a carefully chosen initial distribution ($\rho_{{\rm initial}}$) and it is important to note that this initial distribution is dependent on the parameter $\epsilon$.
>
> > It would greatly help if the authors could add a discussion on why or why not standard (Gaussian) initializations are able to achieve the initialization requirement of Theorem 3.2 / Lemma 3.3.
>
> We would like to emphasize that given our result applies to any width and any data, it does not seem plausible that initializing from the Gaussian distribution will always lead to convergence to the approximate global minima.
>
> Rather, in the (newly numbered) Theorem 3.2 we are able to quantify - possibly for the first time - as to how close to the global minima is it possible to reach parametric in the initialization distribution (which could be chosen to be Gaussian) and the loss function.
>
> Lastly, we would like to emphasize that the explicit expression of the Poincaré constant, for even common distributions, is a distinct and intricate field of research. This complexity is further compounded when dealing with Gibbs measures associated with neural losses, as encountered in our study. Bounding such constants is a challenge that goes beyond the scope of our current work which is about establishing a possibility in optimization theory of deep-learning which had not been recognized earlier but had been highly speculated to be true.
>
> > Another interesting contribution would be to show that for some data distributions, the amount of regularization $\lambda$ required does not hurt performance.
>
> In response to your suggestion, we have expanded our paper to include additional experiments on the MNIST dataset with the same depth-2 neural losses exactly as considered in the main theorems. These supplementary experiments, are detailed in Appendix E, provide further insights into the efficacy of the regularization regime considered in our theory. Notably, our results showcase the remarkable performance achievable by running SGD on this real data set for losses compatible with theory.
>
> And to the best of our knowledge, such performance remains unattainable on these architectures without the incorporation of regularization.
>
> > It is not clear why Theorem 3.2 is stated as a theorem and the closely related Lemma 3.3 which has a broader result is stated as a lemma. It might make more sense to present Lemma 3.3 as a Theorem and Theorem 3.2 as its corollary.
>
> Thanks for this suggestion and we have made the necessary adjustments to make this clear. The newly numbered Theorem 3.2 is now the general guarantee on the quality of the loss value attained parametric in the initial distribution and as a corollary to it Lemma 3.3 now gives the guarantee of convergence to the global minima when the initialization is chosen in a specific way.
>
> > Lemma 3.1 seems a bit out of place and perhaps the readers would need additional context to understand what is the significance of this lemma (hence appearing right before presenting the main theorem) and its relation to the main results.
>
> We have now added some more text before stating Lemma 3.1, so as to help the reader see the context for measuring the Lipschitz smoothness constant.

---

### Review · Reviewer_Aimc · 2023-11-01

**Summary Of Contributions:**

This paper claims to prove the convergence of two layer nets to the global minimum under SGD

**Audience:**

Yes

**Broader Impact Concerns:**

fine

**Claims And Evidence:**

No

**Requested Changes:**

See weakness

**Strengths And Weaknesses:**

I am not quite sure about the results of this work. First of all, there are many typos and nonstandard punctuation in the manuscript. The references are also not in the correct style.

Beside the heavy reliance on the results of Shi et al, (2020), I think the main weakness of this work is its reliance on the almost always incorrect SDE in equation (4), where the noise is assumed to be a full-rank isotropic Gaussian -- which then implies the Gibbs measure, which is also what this work heavily relies on. While this type of noise does correspond to that of SGLD, it does not equal minibatch SGD.

The invariant measure of SGD is far from the gibbs measure, and the noise in the SDE is both parameter dependent (often called the multiplicative noise) and low-rank. In fact, it is very easy to run into loss functions in deep learning for which the SDE in Eq. (4) diverges, whereas actual SGD is convergent. This means that SGD is not really "arbitrarily well approximated" by Eq (4). I think that the authors ought to compare and discuss the results in this work: https://arxiv.org/abs/2308.06671

While the results of the authors might be interesting per se, I do not think they are relevant regarding minibatch SGD. The authors need to rewrite their claims to match their actual contribution

---

> ### Author Response · Authors · 2023-12-18
> **Edits in the Writing Have Been Made Based on The Suggestions**
>
> Thanks a lot for your very careful reading of our manuscript.
> Kindly find below the specific responses that we give for the various important issues that you have raised.
>
> > I think the main weakness of this work is its reliance on the almost always incorrect SDE in equation (4), where the noise is assumed to be a full-rank isotropic Gaussian -- which then implies the Gibbs measure, which is also what this work heavily relies on. While this type of noise does correspond to that of SGLD, it does not equal minibatch SGD.
>
> We would like to bring attention to the fact that the work Shi et al. (https://arxiv.org/abs/2004.06977) in their Proposition 3.5 have already established the conditions of proximity between the standard Stochastic Gradient Descent (SGD) to the Stochastic Differential Equations (SDE) that we consider. So we would like to emphasize that there is no guesswork involved in our choice of the SDE and we are focussed on finding natural neural settings where the conditions stated in their theorems hold -- in particular their Corollary 3.3 and Theorem 3(a) which we directly invoke.
>
> Kindly note that the work of Shi et. al. did not give any explicit M.L. scenario where their theoretical framework is applicable. Further, and most interestingly, the authors there had surmised at the top of their page 9, "some loss functions used for training neural networks might not satisfy this condition". Our work is inspired from this statement and our key contribution lies in bridging this gap by showcasing, that there does exist standard neural loss functions (of  practical relevance) that satisfy the complicated set of criteria that were brought to light by Shi et. al. for their mechanism of global convergence of SGD to apply.
>
> This in turn leads us to bridge an important gap in the theory of deep-learning and that ours is the first-of-its kind proof of classification by neural nets whilst not assuming anything about the size of the nets or the data.
>
> > The invariant measure of SGD is far from the gibbs measure, and the noise in the SDE is both parameter dependent (often called the multiplicative noise) and low-rank. In fact, it is very easy to run into loss functions in deep learning for which the SDE in Eq. (4) diverges, whereas actual SGD is convergent. This means that SGD is not really "arbitrarily well approximated" by Eq (4).
>
> We wholly agree with the reviewer that the invariant measure of Stochastic Gradient Descent (SGD) is not necessarily Gibbs; however, it is crucial to note that this distinction does not impact any stage of the analysis in Shi et al. In particular their Proposition 3.5. is a measure of the gap between the expected values of the loss/objective functions at comparable stages of evolution of the standard SGD and the chosen SDE and it is not a statement about their distributional proximity.
>
> We thank the reviewer for pointing out that our phrase "arbitrarily well approximated" is not a correct representation of what the mathematics says. In our revision, we have carefully aligned the language in our Section 4 to accurately reflect the content of Proposition 3.5 of Shi et. al.
>
> > I think that the authors ought to compare and discuss the results in this work: \href{https://arxiv.org/abs/2308.06671}{https://arxiv.org/abs/2308.06671}
>
> We appreciate the suggestion to compare and discuss the results presented in this very interesting work (\href{https://arxiv.org/abs/2308.06671}{https://arxiv.org/abs/2308.06671}). Firstly, it's crucial to note that our setup does not enjoy rescaling symmetry as used in this paper. Secondly, and most importantly, this paper's results (in particular about the mixing measure of SGD) are about a neural proxy model and none of the claims made have been proven for actual neural nets. For future work, it could be an interesting direction to investigate this possibility.
>
> In sharp contrast to the above reference we work with actual realistic neural net losses and directly prove training for them via SGD, a standard algorithm.

---

### Review · Reviewer_qMSF · 2023-12-06

**Summary Of Contributions:**

This paper studies the global convergence of SGD for logistic loss on two layer neural nets. Authors show a provable convergence of SGD to the global minima with regularization and logistic loss. Authors also prove an exponentially fast convergence rates for continuous time SGD with SoftPlus activation function. An experiment is conducted for different regularizers.

**Audience:**

Yes

**Broader Impact Concerns:**

I have no concerns on the broader impact of this paper.

**Claims And Evidence:**

Yes

**Requested Changes:**

The norm should be given explicitly when it is applied. It will be great if the authors can make some discussion on the question I listed above.

**Strengths And Weaknesses:**

The paper considers an interesting problem and is very well written. The proofs are clear and easy to follow and seem to be correct as far as I checked. Since I am not an expert in this area, I have one question for the authors.

1. The theoretical results of this paper seem to be very dependent on the paper Shi et al. (2020). I hope the authors could introduce this paper Shi et al. (2020) in more details including its background and main results. I also hope the authors could discuss about the main inovation of your paper compared to Shi et al. (2020).

2. It seems that the norm $\left\|| \cdot \right\||_{\mu_s^{-1}}$ is not defined in the paper.

---

> ### Author Response · Authors · 2023-12-18
> **More Background Added About the SDE Used**
>
> We thank you for your kind comments and kindly find our responses below for the two issues you raised.
>
> > The theoretical results of this paper seem to be very dependent on the paper Shi et al. (2020). I hope the authors could introduce this paper Shi et al. (2020) in more details including its background and main results. I also hope the authors could discuss about the main inovation of your paper compared to Shi et al. (2020).
>
> We have addressed this concern by incorporating a more detailed overview to the referenced paper in Appendix D. This additional section provides comprehensive background information and outlines the steps to get the main results of Shi et al. (2020) about the mixing of the SDE under consideration here. We hope this helps to further contextualize our theoretical framework.
>
> With regards to the main results of the paper,  we note that in Shi et al. (2020) the authors did not give any explicit M.L. scenario where their theoretical framework is applicable. Further, and most interestingly, the authors there had surmised at the top of their page 9 ``some loss functions used for training neural networks might not satisfy this condition''. Our
> work is inspired from this and our key contribution lies in bridging this gap by showcasing, that there does exist standard neural loss functions (of practical relevance) such that satisfy the complicated set of criteria that were brought to light by Shi et. al. for their mechanism of global convergence of SGD to apply.
>
> This in turn leads us to bridge an important gap in the theory of deep-learning and that ours is the first-of-its kind proof of classification by neural nets whilst not assuming anything about the size of the nets or the data.
>
> > It seems that the norm $\Vert{\cdot}\Vert_{\mu_s^{-1}}$ is not defined in the paper.
>
> We have now explicitly defined the norm in the statement of Theorem 3.2

---

### Decision · Action_Editor_VAfn · 2024-01-15

**Recommendation:** Accept with minor revision

**Comment:**

In light of the evaluation policy of TMLR (technical correctness and relevance), the paper meets the acceptance criteria. However, there is room for improvement, particularly, the limitation of the work should be clearly stated. Therefore, we recommend the acceptance with minor revisions provided that the authors clarify limitation (see below) and fix typos and citation format (e.g.,"... when it exists Zhang et al. (2018)." should be "...when it exists (Zhang et al., 2018”).

The major concern is a lack of discussion about the strength of regularization. Usually, the regularization coefficient $\lambda$ should be chosen to be small depending on the problem setup (e.g., the number of examples) to achieve a small population error. However, the required condition on $\lambda$ could restrict choosing a small value. Moreover, the condition imposed on the initial distribution is also strong. Given the existing study that demonstrates the learnability of two-layer networks trained with SGD on the logistic loss (e.g., [Ji and Telgarsky (2020)]) under a reasonable condition, the obtained result in the paper appears to be rather restrictive.

**Audience:**

This paper could be of interest to those doing research on the optimization and theory of neural networks.

**Claims And Evidence:**

This study delves into the convergence of SGD for the L2-regularized logistic loss with a two-layer neural network. The authors establish the global convergence of SGD with a small step size by analyzing stochastic differential equations for modeling SGD and validating Poincaré inequality for the associated Gibbs distribution. Furthermore, this work derives the convergence complexity of SGD, considering a carefully selected initial distribution.

**Resubmission Of Major Revision:**

The authors may consider submitting a major revision at a later time.

---

> ### Author Response · Authors · 2024-01-22
> **Thanks**
>
> Thanks a lot for your comments.
>
> Kindly see the revised version where we have included in the conclusion a more detailed discussion of the gaps that remain between our proof and experimental data
>
>  -- and we have also included a more detailed comparison against this work https://openreview.net/forum?id=HygegyrYwH that you referred to.

---

> > ### Author Response · Authors · 2024-02-13
> > **Camera-Ready Version Uploaded**
> >
> > Kindly note that we have now submitted a de-anonymized camera-ready version of the paper,
> >
> > -- which is also accompanied by a video presentation.